# Electrochemical Properties and the Adsorption of Lithium Ions in the Brine of Lithium-Ion Sieves Prepared from Spent Lithium Iron Phosphate Batteries

**Hsing-I Hsiang *** and **Wei-Yu Chen**

Department of Resources Engineering, National Cheng Kung University, Tainan 70101, Taiwan
* Correspondence: hsingi@mail.ncku.edu.tw

**Abstract:** Because used $LiFePO_4$ batteries contain no precious metals, converting the lithium iron phosphate cathode into recycled materials ($Li_2CO_3$, Fe, P) provides no economic benefits. Thus, few researchers are willing to recycle them. As a result, environmental sustainability can be achieved if the cathode material of spent lithium-iron phosphate batteries can be directly reused via electrochemical technology. Lithium iron phosphate films were developed in this study through electrophoretic deposition using spent lithium-iron phosphate cathodes as raw materials to serve as lithium-ion sieves. The lithium iron phosphate films were then coated with a layer of polypyrrole (PPy) conductive polymer to improve the electrochemical properties and the lithium-ion adsorption capacity for brine. Cyclic voltammetry, charge/discharge testing, and an AC impedance test were used to determine the electrochemical properties and lithium-ion adsorption capacity of lithium-ion sieves. The findings indicate that lithium iron phosphate films prepared from spent $LiFePO_4$ cathodes have a high potential as a lithium-ion sieve for electro-sorption from brine.

**Keywords:** spent lithium-ion battery; lithium iron phosphate; lithium-ion sieve; electro-sorption; polypyrrole

## 1. General Introduction

There are three primary cathode chemistries for lithium-ion batteries: lithium nickel manganese cobalt oxide (NMC; specifically, one with 60% nickel, 20% manganese, and 20% cobalt (NMC-622)), lithium nickel cobalt aluminum oxide (NCA), and lithium iron phosphate (LFP). NMC and NCA batteries provide most of the storage capacity in light-duty electric cars, whereas LFP cells are prominent in electric buses. LFP batteries have several advantages over NMC and NCA batteries, including a theoretical capacity of up to $170 \, mAhg^{-1}$, strong high-temperature properties, extended cycle life, cheap material cost, and inexpensive pricing [1]. Furthermore, the safety of lithium-iron phosphate batteries is the highest among existing cathode materials, and it is expected that the demand for and production of lithium-iron phosphate batteries will increase [2].

Yang et al. [3] compared the revenue generated by the same processing technology on recycled batteries of different chemistries and found that LCO chemistry generates the most revenue due to the high price of cobalt; LFP and LMO battery chemistries generate the most negligible revenue due to the low prices of their component chemicals of iron, phosphate, and manganese. As a result, we may conclude that LFP is the least recyclable. However, due to its superior performance, it is progressively replacing other materials as the primary material for lithium-ion batteries [4], and the spent LFP must be investigated for recycling potential.

Lithium reservoirs in brine and seawater are much larger than those in ore. As a result, lithium extraction from brine and seawater is the most crucial method now and in the future. Numerous methods for extracting lithium include evaporative crystallization [5], precipitation [6], solvent extraction [7], and ion exchange [8]. Ion exchange is the most

environmentally friendly and cost-effective method [9]. Selective adsorption has received much attention in recent years, and the lithium adsorbents used are mainly based on $MnO_2$, $TiO_2$, and aluminum hydroxide [9–11]. Although lithium ions can be intercalated in the crystal lattice of the adsorbents, their adsorption capacity varies greatly. The capacity of lithium adsorption is affected by the solution type, the lithium concentration, and the adsorbent composition. Only highly concentrated lithium-containing solutions (>5 mg/L) can achieve a high adsorption capacity (>20 mg/g). Brine water contains Mg, Na, K, Ca, and other elements that may enter the adsorbent's lattice and affect lithium adsorption. From the standpoint of economy and environmental protection, selective adsorption has the advantages of simplicity, better selectivity, higher recovery rate, and less pollution than other techniques, so it has the potential for development. Furthermore, because the amount of lithium resources in seawater is much more significant than those on an inland lake, lithium extraction from seawater is a major trend in the future development of lithium resources. Among these, lithium adsorbent in the ion exchange method is the most cost-effective to extract using a lithium-ion sieve [12–14]. Although it has the advantages of high stability and selectivity, it has some operational drawbacks, such as low actual adsorption capacity, short lifetime, high adsorbent dissolution rate, and slow exchange rate [15,16].

Binders are often added to lithium-ion sieves to provide good mechanical strength and industrial column operation for the granulation process. The adsorption capacity of ion-sieves reduces after granulation due to the covering of active sites, which is the most severe issue with granulation. Furthermore, after the adsorption is complete, the lithium ions are exchanged by acid washing, and in this repeated process, the binder is easily decomposed, resulting in the adsorbent loss [16]. For the granulated-type lithium-ion sieves, the slow adsorption rate is attributed to the spontaneous adsorption of lithium-ion in brine. To improve the exchange efficiency of the lithium-ion sieve, we deposited lithium iron phosphate powder recovered from spent LFP batteries on the titanium mesh substrate with a large specific surface area by an electrophoretic process and then used it as an electrode for lithium-ion electro-sorption. The LFP battery can be charged and discharged by intercalating and de-intercalating lithium ions. The lithium ions in brine water are intercalated in the LFP electrode by adjusting the applied electric field and then placed in the enrichment tank to be de-intercalated so that the $Fe^{2+}/Fe^{3+}$ oxidation-reduction process happens in the LFP structure to concentrate the lithium ions. This LFP electrode offers numerous benefits, including low cost, environmental friendliness, high selectivity, high cycle efficiency, and long service life.

The lithium-ion adsorption is strongly dependent on the conductivity of the electrode. However, the conductivity of lithium-ion sieves of LFP is relatively low. In this study, an electrophoretic deposition process was used to prepare a thin film of LFP with a layer of polypyrrole (PPy) to improve the conductivity of the LFP film. Then the electrochemical properties and lithium-ion electro-sorption capacity of the LFP electrodes were investigated.

## 2. Experimental Procedure

The experiment is divided into three major sections: recovery of LFP cathode, preparation of lithium-ion sieve, and electro-sorption and desorption cycle.

### 2.1. Battery Disassembly and Obtaining LFP Cathode

First, the spent LFP battery was immersed in NaCl solution [17,18], and the battery's residual power was discharged until the voltage fell below 1.5 V. The positive LFP electrode (Figure 1) was separated from the negative electrode (graphite) and electrolyte and then soaked in deionized water for one day. As LFP is a polar material with good compatibility with deionized water, it is easier to scrape the LFP from the aluminum foil. Then LFP cathode was soaked in DMF for 24 h. DMF can dissolve PVDF in the positive electrode to obtain LFP powder. After centrifugal separation and drying, the LFP powder was calcined at 300 °C for 1 h under a nitrogen atmosphere.

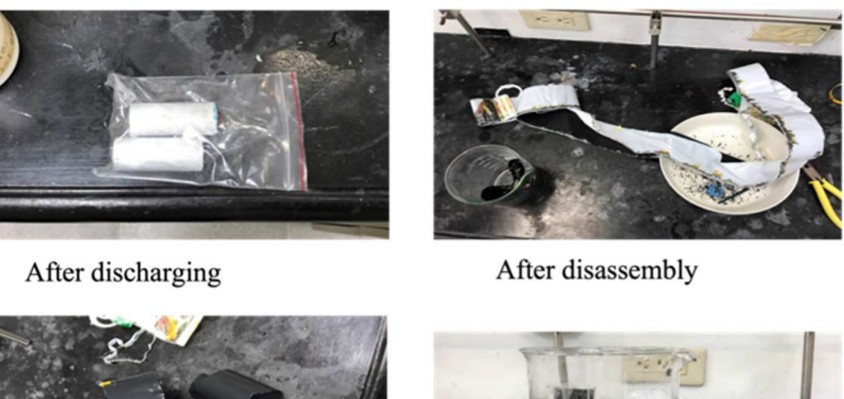

**Figure 1.** Positive LFP electrode separated from the negative electrode (graphite) and electrolyte.

### 2.2. Electrophoretic Deposition of a Lithium-Ion Sieve

An amount of 1.25 g LFP powder was ultrasonically mixed with 25 mL deionized water and 1% poly (4-styrene sulfonic acid) (PSSA) for 40 min to prepare the LFP slurry. A platinum sheet with dimensions of 2 cm $\times$ 1.3 cm was used as the positive electrode and titanium mesh as the negative electrode for electrophoresis. In the electrophoretic deposition process, the titanium mesh or graphite was biased at about +4 V with respect to the reference electrode, the Pt, for 120 s. Following electrophoresis, the lithium-ion sieve was heat-treated at 300 °C for 1 h in a nitrogen atmosphere to strengthen the bond of the LFP film to the substrate.

### 2.3. Electrochemical Synthesis of Polypyrrole

Cyclic voltammetry (CV) was employed for the synthesis of PPy films at a scanning rate of 100 mV/s in −0.2 to 1.02 V vs. Ag/AgCl for 5 cycles using a standard three-electrode cell from 0.1 M 2,6-naphthalenedisulfonic acid disodium salt $C_{10}H_6(SO_3Na)_2$ and 0.1 M pyrrole aqueous solutions [19]. The samples were immersed in double distilled water for 30 min after PPy deposition to remove the pyrrole monomer. A platinum wire served as the counter electrode, and an Ag/AgCl (KCl saturated) electrode was used as the reference electrode.

### 2.4. Electrochemical Characterization

An electrochemical workstation equipped with a potentiostat/galvanostat (Solartron 1287) and an impedance analyzer (Solartron 1260, Bognor Regis, West Sussex, UK) was used to measure and analyze cyclic voltammograms (CV), galvanostatic charge-discharge (GCD) curves, and impedance spectra (EIS). The electrochemical characteristics of the lithium-ion sieve were examined using a three-electrode system in a 1 M LiCl electrolyte, with an Ag/AgCl reference electrode and a Pt counter electrode. The electrolyte was an aqueous 1 M LiCl solution. CV analysis was conducted under the potential window of −0.5 V–0.75 V (vs. Ag/AgCl) and a scan rate of 1 mV s$^{-1}$. GCD analysis was conducted in the potential window of 0.0–0.5 V (vs. Ag/AgCl) and at a current density of 150 mAg$^{-1}$. EIS was carried out in the frequency range from 10 mHz to 15 kHz. A sinusoidal voltage of 5 mV was applied during EIS.

### 2.5. Experiment with Electro-Adsorption and Desorption

The lithium-ion sieve was discharged with a 1 V for 20 h before electro-adsorption and desorption. The electro-adsorption and desorption experiments were carried out with a two-electrode system, with a lithium-ion sieve as the working electrode (used for $Li^+$ adsorption and desorption) and a platinum sheet as the counter electrode. The adsorption solution was simulated brine: LiCl 0.21 M, NaCl 3.3 M, $Na_2SO_4$ 0.172 M, KCl 0.46 M, $MgCl_2$ 0.4 M, $CaCl_2$ 0.0075 M, and $H_3BO_3$ 0.06 M. The desorption solution was 30 mM KCl. The adsorption and desorption times were all 30 min, with 4 cycles. Adsorption was carried out at a voltage of 0.3 V. $Li^+$ was adsorbed by the lithium-ion sieve and converted to $LiFePO_4$. Then, with a voltage of 1 V, $Li^+$ was de-intercalated, and $LiFePO_4$ was re-oxidized to $FePO_4$.

### 3. Results and Discussion

Figure 2 shows the CV result of the $LiFePO_4$ film deposited on the graphite substrate at a sweep rate of 1 mV s$^{-1}$ within a potential window of −0.5–0.75 V (vs. Ag/AgCl). It shows that a pair of redox peaks appear on the CV curve of $LiFePO_4$ in 1 M LiCl aqueous solution, with the oxidation peak near 0.45 V and the reduction peak at 0.2 V. This finding demonstrates that the redox reaction of LFP film during CV characterization is related to the intercalation/de-intercalation of lithium-ion, as well as the viability of $LiFePO_4$ as a lithium-ion sieve. The redox peaks are attributable to the following Faradaic reactions involving [20]:

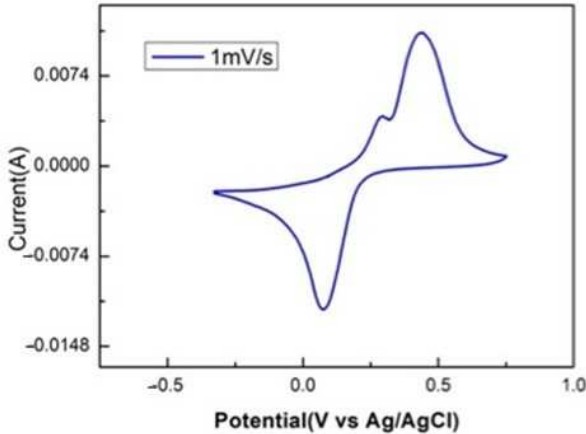

**Figure 2.** CV result of the $LiFePO_4$ film deposited on the graphite substrate at a sweep rate of 1 mV s$^{-1}$ within a potential window of −0.5–0.75 V (vs. Ag/AgCl).

Oxidation:
$$\text{Positive electrode: } LiFePO_4 \rightarrow Li^+ + FePO_4 + e^- \tag{1}$$

$$\text{Negative electrode: } Li^+ + e^- \rightarrow Li \tag{2}$$

Reduction:
$$\text{Positive electrode: } e^- + Li^+ + FePO_4 \rightarrow LiFePO_4 \tag{3}$$

$$\text{Negative electrode: } Li \rightarrow Li^+ + e^- \tag{4}$$

The CV results for the LFP film deposited on different substrates (Ti: titanium mesh and C: graphite) and with or without PPy coating are shown in Figure 3. It demonstrates that peak currents in LFP films with PPy coating were higher than in samples without PPy coating, indicating that the PPy coating promoted the contribution of electron and lithium ions to the redox reaction [21]. The electrochemical polymerization of the PPy film increased the conductivity of the lithium-ion sieve, increasing the peak current and utilization rate of the active material involved in the redox reaction [22,23]. This is because PPy reduced the charge transfer resistance between the $LiFePO_4$ particles and electrolytes, increasing

the proportion of active material used. The peak currents for the LFP films deposited on titanium mesh and graphite cannot be compared due to the different material loadings.

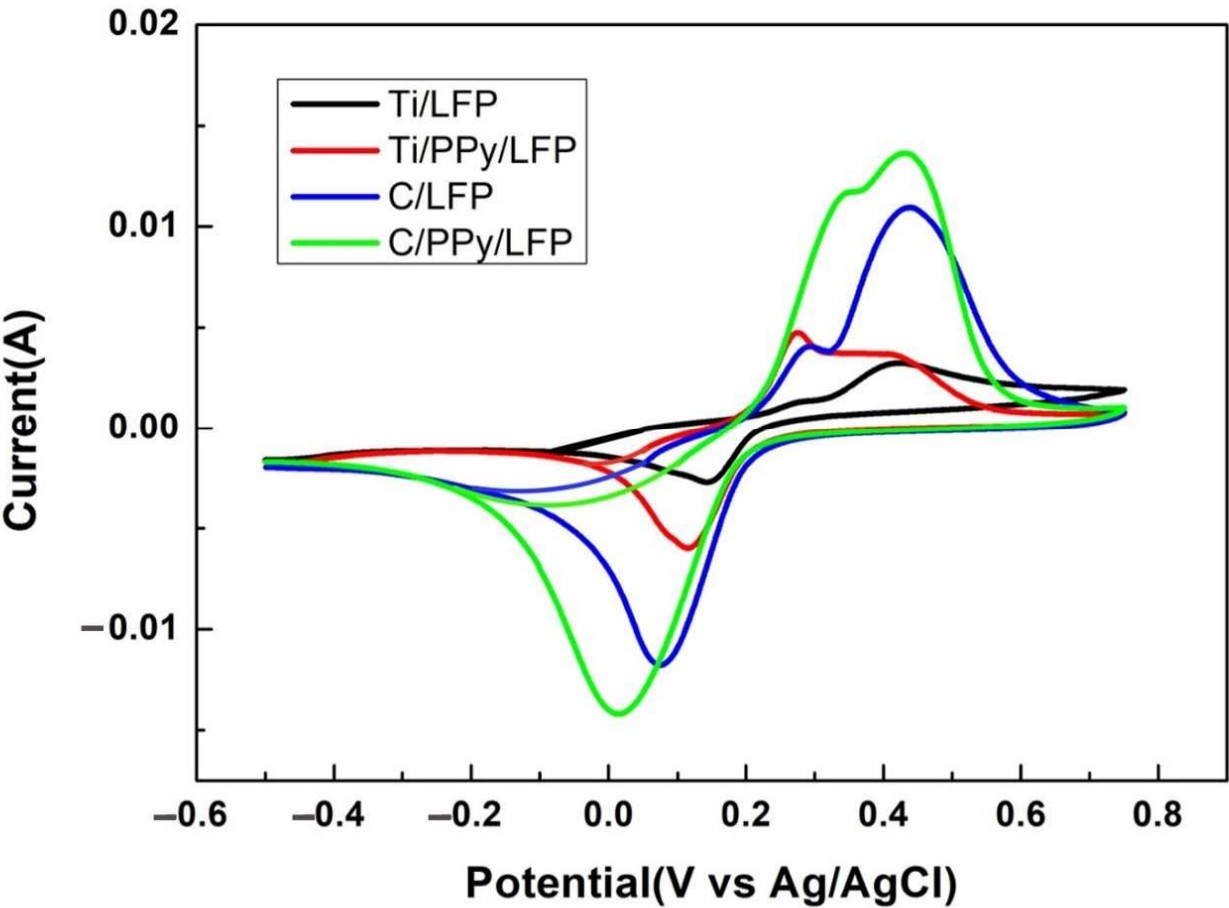

**Figure 3.** CV results for the LFP film deposited on different substrates (Ti: titanium mesh and C: graphite) and with or without PPy coating.

The charging and discharging curves of lithium-ion sieve electrodes with graphite and titanium mesh as substrates are shown in Figure 4. The highest charging level of the $LiFePO_4$ film on the graphite substrate was approximately 0.45 V, which corresponds to the oxidation peak of the CV curve (Figure 2). The discharge platform, on the other hand, represents the reduction reaction. A discharge platform can be seen on the discharge curve of the lithium-ion sieve electrodes. The discharge platform appeared near 0.2 V, which corresponds precisely to the CV curve's reduction peak, indicating that $Fe^{3+}$ in the lithium-ion sieve is continuously reduced to $Fe^{2+}$. The longer the discharge time, the more active materials participate in the reaction. As a result, the amount of active material used can be evaluated by comparing the length of time on the discharge plateau or the discharge capacity of the lithium-ion sieve electrode. PPy-coated electrodes had a more prolonged discharge plateau time than pure LFP film. It implies that PPy can significantly improve active material utilization. Furthermore, the discharge plateau time was longer when titanium mesh was used as the substrate because titanium mesh is a porous material with a larger surface area and more active material in contact with the electrolyte. The capacities of LFP/graphite: 20.66 mAh/g; PPy/LFP/graphite: 27.94 mAh/g; and PPy/LFP/Ti: 29.64 mAh/g demonstrate that PPy-coated LFP film on the titanium mesh had the highest active material utilization efficiency.

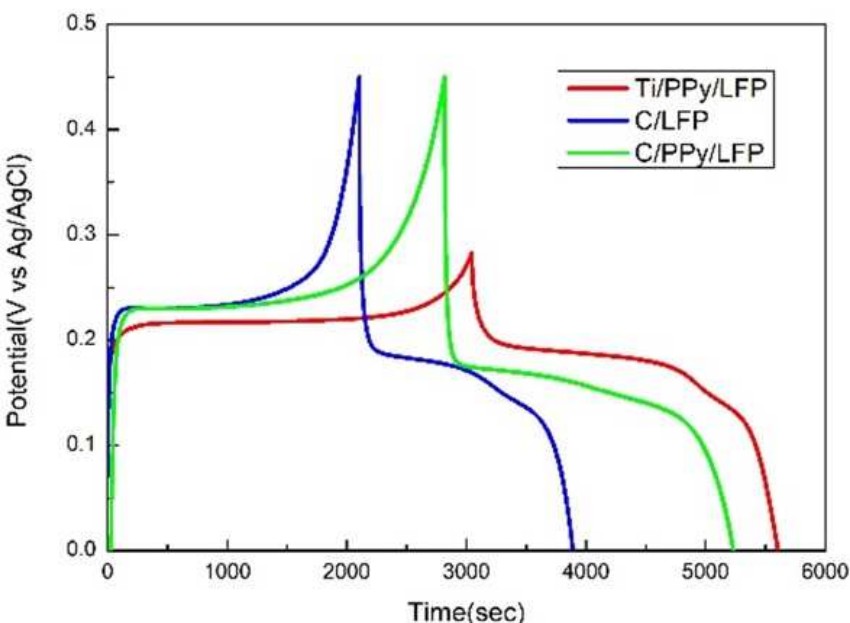

**Figure 4.** Charging and discharging curves of lithium-ion sieve electrodes with graphite and titanium mesh as substrates.

Figure 5 shows the change in capacitance of the lithium-ion sieve electrode with graphite as the substrate after 10 charging-discharging cycles. It has been demonstrated that PPy can significantly reduce charge transfer resistance, allowing more active materials to participate in the charging and discharging process, resulting in a significant increase in capacity from 9.73 mAh/g (without PPy coating) to 30.2 mAh/g.

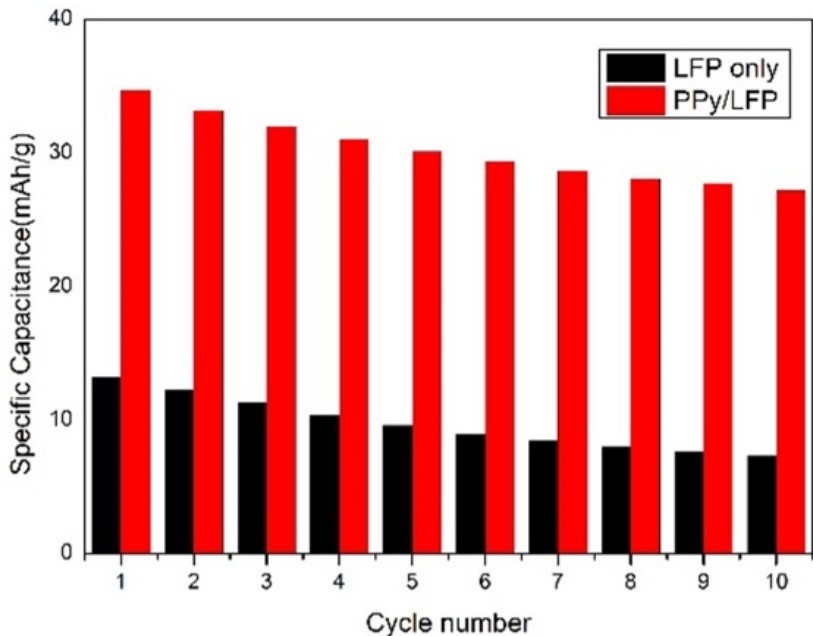

**Figure 5.** Change in capacitance of the lithium-ion sieve electrode with graphite as the substrate after 10 charging-discharging cycles.

Figure 6 shows the SEM microstructure of a lithium-ion sieve electrode on a graphite substrate after 10 charging-discharging cycles. After 10 charging-discharging cycles, the pure LFP electrode had a more significant dissolution loss; however, the physical barrier formed after coating with PPy can reduce the direct contact between the active material

in the electrode and the electrolyte, protecting the crystal structure of the ion sieve and suppressing its disproportionation reaction, thereby slowing the rate of capacity decline. This demonstrates that PPy can inhibit $LiFePO_4$ dissolution loss and thus improve the stability of lithium-ion sieves.

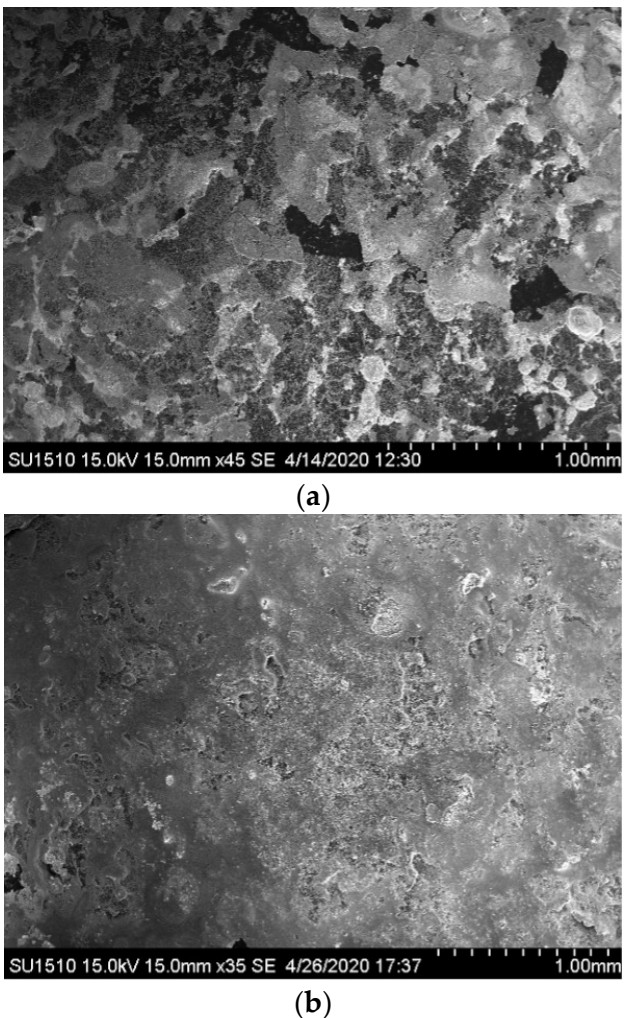

**Figure 6.** SEM microstructures of a lithium-ion sieve electrode on a graphite substrate after 10 charging-discharging cycles (**a**) pure LFP electrode and (**b**) LFP film-coated PPy.

Figure 7 shows the capacity change of a lithium-ion sieve electrode with graphite and titanium mesh as the substrate after 10 charging-discharging cycles. The capacity of the lithium-ion sieve electrode with titanium mesh as the substrate was high, and it remained almost constant after 10 cycles, which was significantly better than the capacity of the lithium-ion sieve electrode with graphite substrate. It has been demonstrated that the porous material and high surface area of titanium mesh allows it to contact more active materials with electrolytes, resulting in a larger capacity; at the same time, the porous structure of titanium mesh allows it to maintain its crystal structure stable when charging and discharging, resulting in longer cycle life. Note that the PPy coating reduced charge transfer resistance, which increased capacity significantly; additionally, because of its physical barrier, it can minimize direct contact between $LiFePO_4$ and electrolyte, protecting the LFP film from disproportionation reaction and thus reducing the rate of capacity decline. To summarize, using titanium mesh as the substrate and the coating of PPy conductive film increased the efficiency and stability of $LiFePO_4$ electrodes significantly.

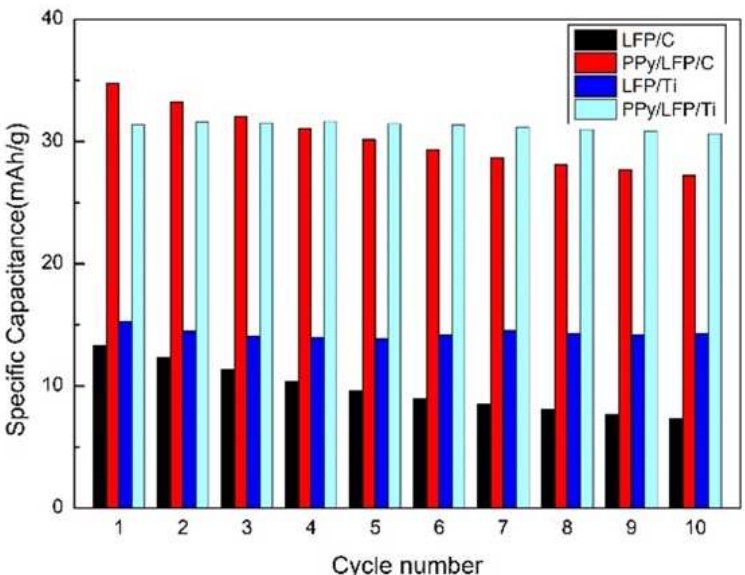

**Figure 7.** Capacity change of a lithium-ion sieve electrode with graphite and titanium mesh as the substrate after 10 charging-discharging cycles.

The effect of PPy conductive film on the AC impedance of LFP film deposited on graphite substrate is shown in Figure 8. The intercept of the high-frequency arc of the LFP film coated with PPy conductive film was greater than that of the electrode not coated with PPy conductive film, but the two were very close. When the slope of the straight line in the low-frequency region was compared, the slope of the electrode without PPy coating was close to 1, indicating that the lithium ions diffused more slowly in the electrode [24]. The slope of the straight line in the low-frequency region was more significant for the electrodes coated with PPy, indicating that PPy can facilitate lithium ion diffusion. This means that the PPy coating on the surface of $LiFePO_4$ allowed lithium ions to pass through and adsorb on the LFP film effectively.

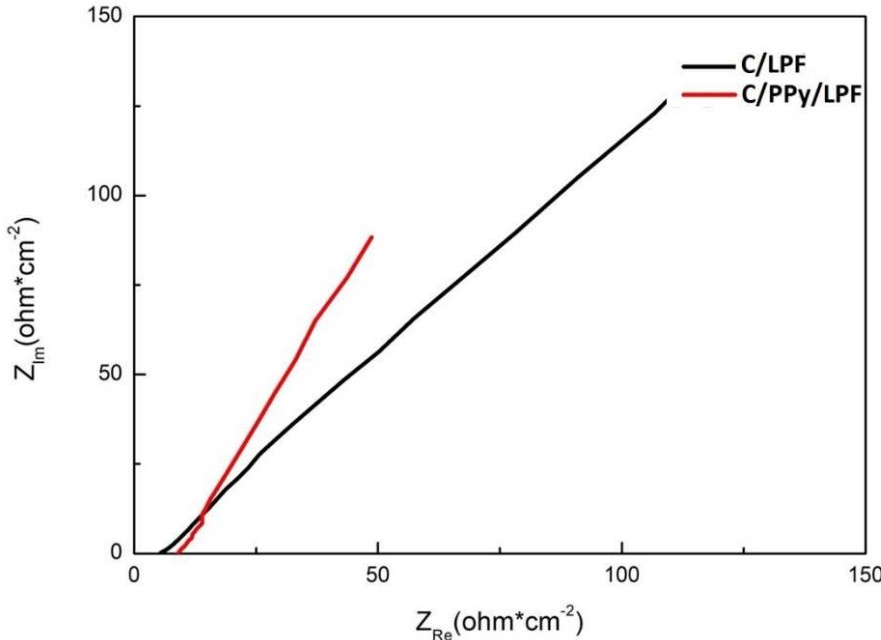

**Figure 8.** AC impedance of LFP film deposited on a graphite substrate.

The number of lithium ions adsorbed in the lithium-ion sieve can be calculated by the following equation:

$$Q_e = (C_0 - C_e) \times \frac{V}{W} \tag{5}$$

$Q_e$: Amount of lithium ions adsorbed. $C_0$, $C_e$: Concentration of lithium ion before and after electro-sorption. $V$: Volume of solution. $W$: Mass of the lithium-ion sieve.

Figure 9 depicts the change in lithium ion concentration in the desorption tank after 4 electro-sorption and desorption cycles for the lithium-ion sieve electrodes. With increasing cycles, the lithium ion concentration in the desorption tank of the lithium-ion sieve electrode on the Ti mesh significantly rises. Table 1 shows the average electro-adsorbed lithium ion amount per cycle calculated using Equation (5). The PPy coating increased the electro-adsorbed lithium ion amount per cycle in LFP films with Ti mesh as the substrate from 0.1 mmole/g to 0.18 mmole/g. The average amount of electro-adsorbed lithium ion per cycle for PPy/LFP/Ti was significantly higher than the value for PPy/LFP/C (0.05 mmole/g). Furthermore, after four electro-sorption and desorption cycles, the adsorption amount of sodium ions still remained constant, indicating that the electro-sorption was highly selective.

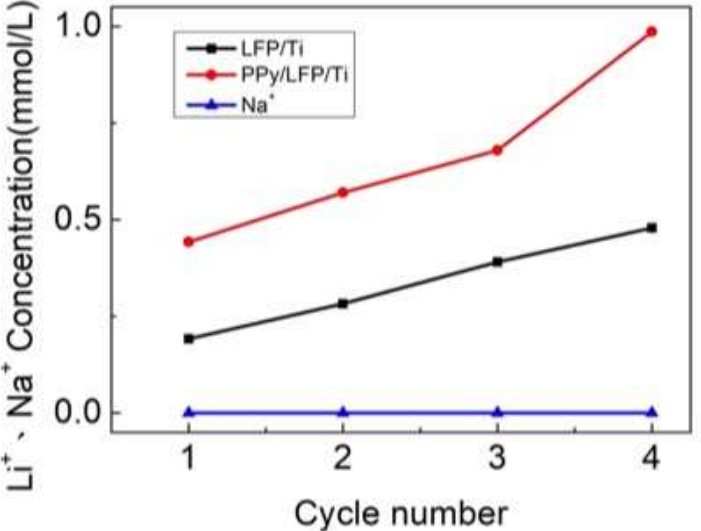

**Figure 9.** Change in lithium ion concentration in the desorption tank after electro-sorption and desorption cycles for the lithium-ion sieve electrodes.

**Table 1.** Average electro-adsorbed lithium ion amount per cycle.

|  | Ti Mesh | | Graphite Substrate | |
| --- | --- | --- | --- | --- |
|  | **LFP** | **LFP/PPy** | **LFP** | **LFP/PPy** |
| Average electro-adsorbed lithium ion (mmole/g) | 0.10 | 0.18 | 0.04 | 0.05 |

Figure 10 depicts the SEM microstructure of the LFP film on a graphite substrate after 10 electro-sorption and desorption cycles. After electro-sorption and desorption cycling, the structure of the LFP film that was not coated with PPy was damaged, but the structure remained intact after coating with PPy. It demonstrates that the PPy conductive film can reduce direct contact between LiFePO$_4$ and electrolyte, thereby protecting the LFP film's crystal structure and extending the life of the lithium-ion electro-sorption and desorption cycle.

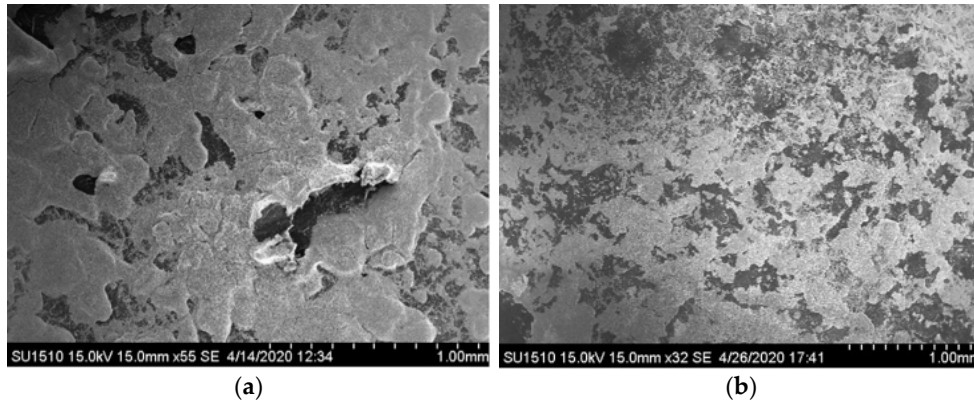

(**a**)  (**b**)

**Figure 10.** SEM microstructures of the LFP film on graphite substrate after 10 electro-sorption and desorption cycles (**a**) pure LFP electrode and (**b**) LFP film coated PPy.

Figure 11 depicts the XRD patterns of the lithium ion sieve electrode after complete discharge (1000 min of delithification) and at various lithium adsorption times. The $LiFePO_4$ phase was nearly converted to the $FePO_4$ phase after a complete discharge. Then the content of the $LiFePO_4$ phase gradually increased with increasing lithium adsorption times. These results indicate that the lithium ions can be intercalated and de-intercalated into the lithium ion sieve electrode reversibly.

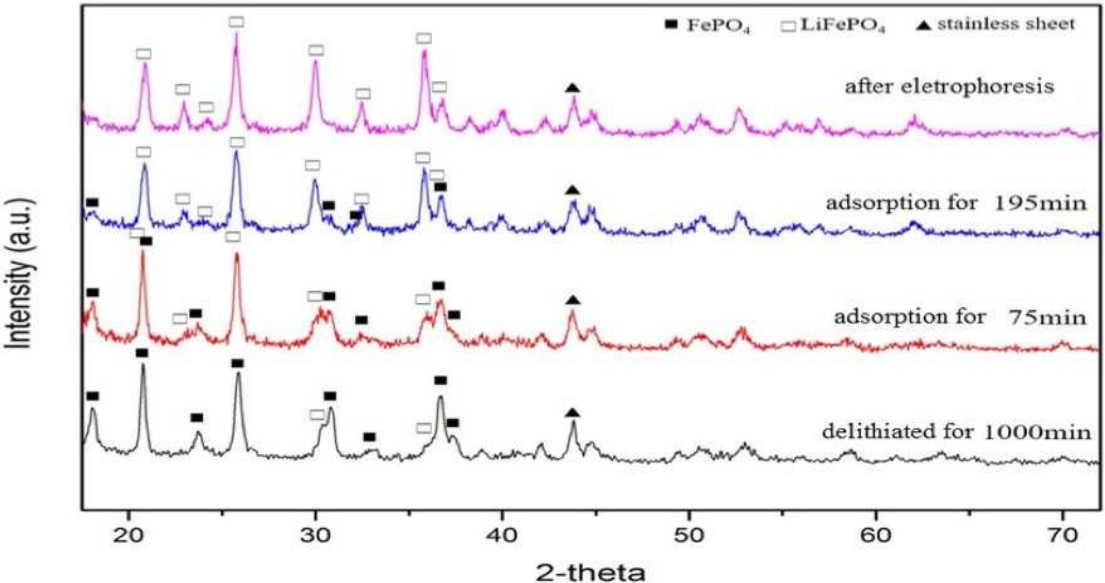

**Figure 11.** XRD patterns of the lithium ion sieve electrode after complete discharge (1000 min of delithification) and at various lithium adsorption times.

## 4. Conclusions

This study successfully developed lithium iron phosphate films by electrophoretic deposition using spent lithium-iron phosphate cathodes as raw materials to serve as lithium-ion sieves. The electrochemical properties and lithium ion adsorption for brine were improved by coating the surface of the lithium iron phosphate film with a layer of PPy conductive polymer. PPy coating can reduce the charge transfer resistance of the LFP film and electrolyte, allowing more active materials to participate in the charging and discharging process. In comparison to the graphite substrate, the titanium mesh was used as the substrate of the lithium ion sieve electrode, which had more active material in contact with the electrolyte and a higher electric capacity; meanwhile, the titanium mesh's

porous structure can maintain the stability of its crystal structure during charging and discharging, resulting in a longer cycle life. The amount of electrosorbed lithium ions in LFP films increased with increasing electro-sorption and desorption cycles, while sodium ions remained almost constant, indicating its great potential as an electro-sorption lithium ion sieve.

**Author Contributions:** Conceptualization, H.-I.H.; Data curation, W.-Y.C.; Formal analysis, W.-Y.C.; Funding acquisition, H.-I.H.; Investigation, H.-I.H. and W.-Y.C.; Methodology, H.-I.H. and W.-Y.C.; Resources, H.-I.H.; Software, H.-I.H. and W.-Y.C.; Validation, H.-I.H. and W.-Y.C.; Writing—original draft, H.-I.H.; Writing—review and editing, H.-I.H. All authors have read and agreed to the published version of the manuscript.

**Funding:** This research received no external funding.

**Institutional Review Board Statement:** Not applicable.

**Informed Consent Statement:** Not applicable.

**Data Availability Statement:** Not applicable.

**Conflicts of Interest:** The authors declare no conflict of interest.

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
