# Peer review of "Electrochemical Properties and the Adsorption of Lithium Ions in the Brine of Lithium-Ion Sieves Prepared from Spent Lithium Iron Phosphate Batteries"

_sustainability, doi:10.3390/su142316235_

Round 1

Reviewer 1 Report

Dear Author,

The current article aims to study electrochemical properties and the adsorption of lithium ions in the brine of lithium-ion sieves prepared from spent lithium iron phosphate batteries developed by a sustainable approach. This article can be accepted after major corrections.

Line no.13 used in place of spent

Line no. 14 few people not good used researchers

Line no. 34 LFP abbreviation not clear

Fig. 2 CV must be complete cycle and used 2nd run or take measurement after few minutes

Fig. 8 AC impedance of LFP graphite substrate in y axis –Zimg , scale should be positive.

Author Response

As the attached file.

Reviewer 2 Report

Authors reported an interesting work on usage of cathode for Lithium separation from brine. However, it can be improved for better readers understanding.

Why authors immersed in NaCl solution?

Why was electrophoresis chosen? Lithium can be separation through electrodeposition?

Authors should also provide lithium deposited LFP electrode

There is no XRD analysis given in the manuscript. It would be better to perform XRD analysis

Author Response

as the attached file.

Reviewer 3 Report

After fully review this manuscript, there are some places need to improve to meet the standard of Sustainability. So, I recommend it publication after major revision. The suggestions have been listed as follows:

1.      The references in this manuscript are too few.

2.      In Figure 3, the CV results for the LFP film deposited on different substrates (Ti: titanium mesh and C: graphite) and with or without PPy coating have been discussed. And the authors concluded the presence of PPy coating promoted the contribution of electron and lithium ion to the redox reaction. However, the factor of different substrates hasn’t been discussed. Moreover, the redox peaks of the CV curves should be also clarified. Especially, the intensity oxidation peaks for Ti/LFP and Ti/PPY/LFP are different.

3.      There are some careless typo errors such as “The PPy coating increased the electro-adsorbed lithium-ion amount per cycle in LFP films with Ni mesh as the substrate…”, It should be Ti,  “…the adsorption amount of sodium ion still remained constant, indicating…” It should be lithium, rather than sodium. The Y-axis in Figure 9 should also be modified.

4.      The inset caption in Figure 8 should be C/LFP and C/PPy/LFP. Please check through the manuscript and normalize these abbreviations.

5.      The contents of Figure 9 are not consistent with the Table 1. Please check and modify it.

6. The format of conclusion is suggested to be modified. This is a scientific report conclusion, rather than experimental conclusion.

Author Response

as the attached file.

Round 2

Reviewer 1 Report

Dear Author,

This article is suitable for publication in present form

In  EIS figure y axis legend must be like this   -Zimag

Reviewer 2 Report

Manuscript can be accepted for publication

Reviewer 3 Report

The authors response the issues well.